# Use of Instagram as a Resource for the Adoption of Behaviors Related to Health and Well-Being of Young College Students: Associations between Use Profile and Sociodemographic Variables—A Cross-Sectional Study

Kaline Pessoa [1,2,*], Cícero Luciano Alves Costa [2], Ana Cláudia Coelho [3], Ana Bastos [1,4] and Isilda Rodrigues [1,4]

1. School of Human and Social Sciences, University of Trás-os-Montes and Alto Douro (UTAD), 5000-801 Vila Real, Portugal
2. Federal Institute of Education, Science and Technology of Ceará, Limoeiro do Norte 62930-000, CE, Brazil
3. Associate Laboratory for Animal and Veterinary Sciences (AL4AnimalS), Animal and Veterinary Research Center (CECAV), University of Trás-os-Montes and Alto Douro (UTAD), 5000-801 Vila Real, Portugal
4. Centre for Research and Intervention in Education (CIIE), Faculty of Psychology and Education Sciences, University of Porto, 4099-002 Porto, Portugal

* Correspondence: kaligia.tc@hotmail.com; Tel.: +55-84-998693848

**Abstract:** The use of Instagram and content from digital influencers to gain information and adopt behaviors related to health and well-being may be associated with sociodemographic variables. Few studies have been conducted in different contexts regarding the use of Instagram to obtain information about health and well-being and its relationship with sociodemographic variables. A descriptive cross-sectional study was performed with a convenience sample of the population of students attending a degree course in physical education, in the northeast region of Brazil, to assess the prevalence of Instagram use as a resource for the adoption of behaviors related to health and well-being, as well as to understand the associations between use profiles and sociodemographic variables. An online validated questionnaire was completed by 162 students from March to June 2021. Descriptive statistics and analysis of artificial networks were used. Results indicate the profile of using Instagram to obtain information about health and well-being is impacted by sociodemographic variables such as gender, age, monthly income, and the semester the student is attending. Specifically, although men adopt health and well-being behaviors more frequently, the relationship between variables such as age and monthly income and the variables of the Instagram use profile is stronger among women. However, the adoption of behaviors and the belief in their contribution to self-care establish a strong relationship among both genders. It is concluded that sociodemographic variables can contribute to a better understanding of the use of Instagram to adopt behaviors related to health and well-being.

**Keywords:** health; well-being; digital influencer; Instagram; college students

## 1. Introduction

Instagram is an attractive and interactive social network (SN) intensely used by mostly young users [1,2] who engage with posts and diverse content published largely by digital influencers (DIs) and is the preferred SN platform of DIs [3,4] to disseminate diverse content, including on wellness, health, and healthy lifestyles.

Digital influencers (DIs) can be considered opinion leaders active in a niche market, attracting users for whom their opinions are considered expert and important [3,5,6]. These can range from celebrities with thousands of followers to micro-influencers, with smaller numbers of followers and higher engagement, and people with experience and expertise in a particular subject [7].

Influence marketing invests heavily in DIs because they inspire trust, are dynamic, are specialized in certain subjects, have credibility, carry out pioneerism, and inspire users by publicizing their lifestyles [8]. As they are ordinary people with social relevance in their networks, DIs create an environment closer to the public [4,9]. Besides the number of followers, they can be classified according to the motivations to communicate, the communication platform used, and the type of activity [3].

Added to this, the social network Instagram stands out for enabling the transmission of a positive experience about the consumption of products and services through images usually edited and formulated to draw attention, with less emphasis on the text [10]. Its focus on the production, editing, dissemination, and sharing of images [8] makes Instagram the most used social network for the activities of influencers [3] who seek to evolve socially in their online spaces by producing creative and innovative content, focusing on maintaining relationships and engagement for consumption [8].

Lifestyle influencing is a specific niche, where influencers tend to attract the public more strongly, especially young people. Fitspiration, for example, is one of the phenomena of dissemination of images in social networks with the aim of inspiring people to adhere to diet and exercise regimes for a healthy life [11].

However, this movement facilitates an environment of addiction due to flashy posts and content based on the exposure of modeled bodies and the use of sponsoring brands, as well as the promotion of diets and exercise associated with the seeking of a body standard [12]. The appealing use of body image, emphasizing sexuality and youthfulness achieved using certain products and submission to certain routines, tends to contribute to reaffirming "a lifestyle that should inspire followers to seek body transformation through consumer practices" [4].

These aspects can be further reinforced by celebrities and sub-celebrities of digital influence during the dissemination of images and videos of their fitness routines, being associated with thinness and curvy bodies by the users who consume them [13]. Other modes of influence are automatic content sources, generated by users' preferences, content with inappropriate health information, peer influence, and interaction through "likes", which function as means of social affirmation [14]. Added to all this is the visual appeal of Instagram and the speed of content dissemination [1].

With the use of digital resources to inform oneself about health and the tendency to search for them on the Internet [15], it becomes necessary to observe the dissemination of this type of content on social networks. Depending on how the interaction with the content occurs and how it is disseminated, it can generate serious damage in those who view it. However, when the quality of information is guaranteed, the broad access offered by social networks can benefit various types of populations and needs [16]. For example, information sharing is a positive and cost-effective feature of SNs, making it fundamental and current in helping health issues by effectively disseminating messages and countering negative views around health promotion [17]. Social networks are also popular digital resources for information about health conditions, including initiatives such as the recruitment of groups with specific conditions and interest in information about their conditions, contributing to their search for health and quality of life and becoming important and efficient tools in public engagement for health promotion [18].

These digital spaces are already being used to assist health professionals and to disseminate information that contributes to health promotion work regarding emerging public health issues. In the ongoing pandemic, the important role of Instagram in disseminating behavioral measures to prevent SARS-CoV-19 (COVID-19) contagion prior to the development of drugs and vaccines is highlighted, as dissemination via SNs increases the chances of various audiences reading such information [19].

The use of these digital resources for health is also identified in research involving the high rate of intervention programs for weight loss behaviors associated with Facebook, Twitter, and Instagram [20], and can positively influence changes in eating and exercise behavior [21]. The information available has the potential to contribute to individuals'

health knowledge, aiming to find health problems in themselves or others from information found on the Internet [22].

In Brazil, data indicate the use of Instagram and other social networks as an important tool for the dissemination of health information through the sharing of images and videos and the constant interaction between professionals and students of the area to address various health issues, as well as to encourage the development of healthy habits and also as a measure to promote health education through popular and expanded access to social networks [23–28].

Other data indicate that active use of SNs such as Instagram to inform oneself about health is positively associated with adolescent and youth subjective well-being [29,30] and motivation for changes in eating and exercise habits [1,11,21,31], through equal access to varied health information [32] and the interactivity enabled by the varied tools of this medium [21].

Social support and peer influence are also perceived as positive factors, enabling health engagement in adolescents and young adults [1] and positively mediating active Instagram use [33], as well as positively impacting mood and life satisfaction issues [29].

However, as noted, the use of Instagram can become equally detrimental to health promotion campaigns, especially for youth and young adults who are increasingly connected [32]. Instagram and Facebook have been noted as the social networks that most influence unhealthy behaviors, such as by advertisements for some types of foods and practices that encourage risky eating behaviors [1]. Research identifies that most of the content searched for refers to diets, exercise, body image, food displays (their classification into healthy or unhealthy), and healthy lifestyles [14,32], which may contribute to inappropriate notions about health and wellness.

Research on young people's consumption of health information on social media points out that despite their familiarity with the spaces, they are not sufficiently prepared to use SNs with respect to health content, mainly because at certain times the medium controls them more than the other way around, since the very interactivity of the medium and young people's modes of use shape the type of health information that reaches them [14].

The damage is detected as passive use, high investment (of time, frequency of use), and ostracism in SNs that have been identified as negative factors related to subjective well-being, causing depression, moodiness, body and life dissatisfaction, and low self-esteem [29]. Other results point to the risk of SN addiction. During the pandemic, for example, there was an increase in SN consumption, especially Instagram, with significant results for addiction [34].

Other effects are pointed out mainly in young women (14 to 35 years) [35] whose consumption behavior of health-oriented content is associated with the search for SN profiles with a focus on body image and appearance, denoting a significant relationship between the use of Instagram and body dissatisfaction, the concept of thinness, and low self-esteem. However, for both women and men, the ways in which healthy lifestyles are publicized tend to become an obsessive ideal to be followed, implicated in self-image, increased levels of anxiousness, and impact on self-esteem [32].

These factors, added to the constant exposure to advertising appeals on Instagram, can have negative consequences for wellness and health [1], and contribute to the spread of harmful lifestyles that, in a free and fun method of promotion, increase youth and adult engagement in their consumption [32].

Lifestyle is defined by the Glossary of Terms of the World Health Organization as the set of habits that are influenced by the socialization process, such as substance consumption, exercise habits, and diet, having fundamental implications for health and being one of the factors for well-being [36].

Concerning health, understanding it as part of the subject's integrality fits well, with several factors influencing their health status including, among them, the lifestyle and behaviors the individual [37]. In addition, we emphasize the dynamic character of health, understood as the result of the adaptation and proper functioning of the individual ac-

cording to the environment, and not the illusory search for perfection. This aspect is more consistent with the notion of adaptation of individuals to the digital environment, of networked communication, and the intense flow of information present in SNs, often bringing information and ways taken as an ideal way to live [37].

In turn, individuals' behaviors are related to the determinants of health and refer to the subjects' choices and lifestyles, but also, and mainly, to the social, political, and environmental conditions that enable and facilitate these behaviors [37].

Regarding well-being, this can be understood as "the harmonious integration between mental, physical, spiritual and emotional components" [38], resulting from the perception and self-assessment of the subject. Well-being represents a dynamic balance between the subject's physical, mental, and social well-being, "a lifestyle that recognizes the importance of nutrition, physical fitness, stress reduction, and personal responsibility" [36].

Taking these issues into account, we observe that young people's uses and preferences for Instagram and the presence DIs imply both positive and negative consequences on users' behaviors and deserve to be highlighted in investigations that seek to understand these facts from various perspectives [8,9].

According to research on the high consumption of content regarding diets, exercise, body image, food, and healthy lifestyles [14,32] and the consequences for young people previously pointed out, we sought to understand the adoption behaviors regarding this content by the research participants, understanding them as behaviors focused on health and wellness and disclosed by the DIs on their Instagram pages, and investigating the relationship with gender, age, education (course semester), and monthly income.

We understand that investigations should seek to understand the sociodemographic differences that influence and characterize the SN use behaviors of young people, being necessary to explore such aspects in varied contexts [39,40], and justifying the preparation of this work. Previous research [41] indicates that gender, age, and formal education correlate with the consumption of fake news disseminated by online articles in SNs. Data point [42] to the importance of understanding SN types separately and comparing usage behaviors regarding the same variables used in the present research, such as gender, monthly income, education, and age.

We, therefore, investigated a group of university students from the northeast region of Brazil, in order to contribute to clarifying the research gap concerning the differences in preferences on the use of SNs such as Instagram [12] and the scarcity of research in Brazil, since most is carried out in other countries [26].

University students are understood as an intensive SN consumer public, finding themselves in a hyperconnected universe [34,43,44]. Studies point out both the vulnerability of this population to addiction and anxiety from thoughtless use [45,46] and consider this population as avid consumers of social networks and as potential learning and information tools [34,43,44].

Since 2010, the number of investigations about students' SN usage behavior has been increasing, with the USA, Asia, and Europe leading the way in publications on college students [47], but there is still a need to focus on the characteristics of social networks and the audience.

Based on these aspects, we sought to understand the prevalence of the use of Instagram as a resource for the adoption of behaviors related to the health and well-being of a group of young Brazilian college students and its associations with the user profile and sociodemographic variables. As central objectives, we aimed to analyze the prevalence of use of Instagram as a resource for the adoption of behaviors related to the health and well-being of a group of young college students and understand the associations between the usage profile and sociodemographic variables.

## 2. Materials and Methods

### 2.1. Type of Study and Sample

A descriptive cross-sectional study was performed with a convenience sample of a population of students (*n* = 162) attending a degree course in physical education, in the northeast region of Brazil, to assess the prevalence of Instagram use as a resource for the adoption of behaviors related to health and well-being, as well as to understand the associations between user profiles and sociodemographic variables. By the time of the survey, 234 students were officially enrolled in the course.

The sample size for this study was calculated according to the formula of survey sample size calculation [48] assuming a 10% expected prevalence of the use of Instagram as a resource for the adoption of behaviors related to health and well-being, therefore 138 participants were required. To allow for a 10% non-response rate, the lowest required sample size was then 155 participants. In order to include as many students as possible, a sample of 162 students was used.

Through social networks and during classes in the online modality, we invited the enrolled students by sending a message with the research topic and objectives, and a brief justification of the work. After this process, we reached the required number of 162 students.

The choice of online media contributed to speeding up the investigative process and has been recommended for its speed between planning and publication of results, which is a relevant aspect in crisis contexts [49]. In addition, the teachers of the course contributed to the dissemination of the invitation, so that we could reach a larger portion of students, considering the pandemic period and the difficulty in accessing online classes, besides the massive dropout of students from the course in this period.

As inclusion criteria, we stated that participants should be regularly enrolled in the course, be 18 years old or older, and use the Instagram social network. Therefore, the goal of the application was to include as many students as possible in order to obtain completed forms. In this sense, this sample type is non-probabilistic, due to seeking representativeness of a population according to its characteristics and the researchers' choice. In this type of sample, "the choice of elements does not depend on probability, but on causes related to the characteristics of the research or who does the sample" [50].

### 2.2. Instrument

For the present study, we designed an online form, with content validated by a panel of experts. For the formulation of the "questionnaire" instrument (see Supplementary Materials), we considered the variability of closed questions, which offer possibilities of dichotomous answers and/or several options for participants to choose from (and their variations) [50].

As a way to minimize these possible flaws of the technique, we used a group of strategies for validating the questions and categories of options and possible answers of the instrument in order to achieve clarity and conciseness and be free of value judgments [50].

To accomplish this, we submitted the instrument to 3 groups: colleagues; experts in the fields of study; and some students who represented characteristics of the target public. Considering this, we could avoid some problems present in this type of technique: adequacy of objectives; inclusion of obvious questions during the elaboration process; and familiarization with questions that are complicated to understand. This type of procedure allowed us to verify the clarity of the instruments, the answers, questions, scales, and content of the answers [50–53].

The procedures for developing and validating the questionnaire are described below:

Formulation of draft questions and answer options with pre-defined categories: we used a basis referring to health, digital influence, and research methodology [6,50,54–59], in order to cover the theoretical and methodological aspects regarding the creation and validation of the questionnaire.

Evaluation and validation steps by a panel of experts, or judges [60]: with the first version ready, a panel of professors was invited to evaluate the instrument. Thus, the

questionnaire was initially evaluated by doctoral professors in the areas of research methodologies, educational technology, and health education, with the appropriate adjustments requested and made. The choice criteria for the experts were: availability of time for the evaluation and working in one of the specific areas of the research theme.

Peer validation and pre-testing: we chose to send the versions approved by the expert panels to two groups: the first was of professors with experience in quantitative health research, in the area of educational technologies and research in health, education, and physical education.

The selection criteria were: teachers available to evaluate and work in the thematic areas of the research and quantitative research methodology. After observing the suggestions and making the necessary modifications, we proceeded to the last validation stage with the pre-test in a public with characteristics close to those of the original sample.

The second group was chosen based on the characteristics of the original sample. The selection criteria for the pre-test stage were: undergraduate classes; availability of time and space for the explanation of the objectives and theme of the study and application of the questionnaire; availability of students to offer feedback on the content of the questionnaire.

Thus, two classes with a total of 47 students answered the questionnaire, and the application followed the same criteria that were used in the original application. During this stage, we asked the students to evaluate the instrument in terms of understanding the questions; writing; understanding the content; and any other modifications they felt were necessary, all of which were noted and considered.

Finally, the final version of the questionnaire was created in the Google Forms platform, following the indications of the Brazilian Ethics Committee (in addition to other appropriate techniques for its formulation), including an informed consent form (TCLE) (a document required by the Committee) so that it could be seen by the students, with the purpose of confirming their majority. All the changes suggested and made to the questionnaire did not significantly alter its initial meaning. The modifications were in the scope of adjusting some concepts, organizing the questions, including/excluding alternatives that caused doubts, excluding or altering identical questions or questions with the same meaning, revising questions and alternatives with double meaning, and technical suggestions related to the creation of the online form. The final version of the questionnaire had 5 sections:

1. Contextualization of the research: elucidation of the objectives and main theme (objective: briefly describe the research).
2. Informed consent form: explanation of all the procedures regarding data collection, participants' rights, and researchers' duties (objective: to obtain students' consent to conduct the research).
3. Characterization of the sample: information about the sociodemographic questions of the students (age, gender, education, monthly income, address) (objective: to identify the sociodemographic data of the sample) from 10 questions.
4. Information about the relationship between students' formative context, Instagram use, and search for digital influencers (objective: to discuss the relationship between the formative context of the investigated subjects and the search for health and wellness content on Instagram) from 12 questions.
5. Information about the profile of Instagram use and the search for digital influencers (objective: to investigate whether the types of content produced by digital influencers on the social network Instagram are configured as promoters of health education) from 19 questions.

The original questionnaire was applied to all classes of the physical education undergraduate course at a federal institute of education located in the northeast region of Brazil, between the months of March and June 2021.

The data of the variables for this work were obtained from the following questions and answers: characterization of the sample (with the options: between 18 and 20 years; between 21 and 25 years; between 26 and 30 years; and above 31 years); gender (with the options: female; male; other); monthly income (the monthly income in the year of the survey was

BRL 1212.) measured by the amount of income received, divided by the number of people in the household (with the options: up to 1 minimum wage; between 1 and 3 minimum wages; between 3 and 5 minimum wages; more than 5 minimum wages; n minimum wages); in which semester of the physical education course the participants are enrolled (with the response options from 1st to 8th semester). All answer options were unique.

In the 5th session, about the use of Instagram and the search for digital influencers, the following questions and options were used: About health, do you usually follow the Digital Influencers that create these contents? (options for single answers: yes; no); How often do you use Instagram and the content disclosed by Digital Influencers to get informed about health and well-being? (options for single answers: always; frequently; rarely; never); Do you adopt the behavior about health and well-being disclosed on Instagram of the Digital Influencers for your life? (single answer options: yes; no); Do you consider that the Digital Influencers you follow on Instagram present a pattern of health achievable by all who follow their suggestions and/or adopt their behaviors? (single answer options: yes; no); Do you consider that the Digital Influencers you follow on Instagram present a pattern of body achievable by all who follow their suggestions and/or adopt their behaviors? (single answer options: yes; no); Do you consider that the contents disclosed by Digital Influencers dealing with health and wellness contribute to your self-care? (single answer options: always; often; rarely; never).

*2.3. Data Analysis*

For the analyses, descriptive statistics were used to compare the frequency of distribution of the adoption of health and well-being behaviors promoted by DIs, according to gender, age, and monthly income.

To verify the association between the different variables studied in relation to gender (male and female), age (up to 25 years and over 25 years), and income (up to 1 minimum wage, 1 to 3, 3 to 5, >5, and no income), the chi-square test and Fisher's exact test were used. The latter was used when the expected frequency was less than 5. The significance level (*p*) was set at 5%. We verified the possibility of the association of these variables with some questions regarding the profile of use and behavior of following digital influencers.

To verify associations between the Instagram use profile for the adoption of behaviors related to health and well-being, monthly income, age group, and semester attended by the college students according to gender, we applied network analysis to generate a graphical representation of the relationships between variables. This type of analysis has been recommended to infer relationships between variables that explain the emergence of complex phenomena [61]. Human behavior, like countless natural and social phenomena, is not characterized by linearity and cause-and-effect relationships, but rather by complexity and chaos [62]. Moreover, complex systems are explained by the interaction between the different components of the system, and not by the sum of its parts [63]. Usual analyses such as linear regressions would not be suitable to analyze this phenomenon, while the topographic analysis of the networks makes it possible to more effectively visualize the complexity inherent in the relationships among the variables studied.

Thus, the analysis of artificial neural networks emerges as an interesting alternative to examine behaviors that are not yet well known, such as the topic of the current study. It provides a graphical analysis that allows us to visualize a topological space of the relationships between the different variables that make up the system. In other words, it shows a pattern of dynamic relationships between system variables.

In the graphical analysis, each variable is considered a node and the lines connecting the nodes are the synapses, which also characterize the weight of the relationships between the nodes [64]. Blue lines represent positive associations between nodes and red lines indicate negative associations. The thickness and intensity of the colors represent the weight of the associations [65].

The measures of centrality, betweenness, closeness, and strength, were reported. These measures are important to understand the role of each variable within the system [48]. Be-

tweenness measures the ability of a node to act as a mediator between other nodes, influencing peers in the network; closeness evaluates the distance of a node to the other nodes in the network; and strength refers to the sum of the weights of the trajectories that connect the nodes [66–68]. Data analyses were performed using the software JASP 0.11.1 (2019).

*2.4. Ethical Aspects*

This research was approved by ethics committees of Portugal and Brazil, with all data being collected only after approval, under the opinion Doc89-CE-UTAD-2020 (Ethics Committee of the University of Trás-os-Montes and Alto Douro (CE-UTAD)) and Consubstantiated Opinion of the CEP, under the reference CAAE 42856720.0.0000.5294, by the University of the State of Rio Grande do Norte, Brazil.

**3. Results**

For the presentation of the results, we will follow the order of the chosen methods of analysis: firstly, we will describe the results obtained from the frequency distribution regarding the adoption of behaviors related to health and well-being by students and, subsequently, we will present the results according to the network analysis, to discuss the associations between the Instagram use profile and the adoption of behaviors related to health and well-being.

*3.1. Frequency Distribution for the Adoption of Health and Well-Being Behaviors Promoted by DIs (Node 6)*

Table 1 presents a descriptive analysis about the adoption of health and well-being behaviors promoted by DIs (node 6). It is possible to see that percentages of adoption of health and well-being behaviors, promoted on Instagram by DIs, are higher in male college students, although there is no significance difference.

**Table 1.** Frequency distribution for the prevalence of adoption of health and well-being behaviors promoted by DIs, according to gender, age group, and monthly income ($n$ = 162).

| | Gender | | | |
| --- | --- | --- | --- | --- |
| | Female | Male | | *p*-Value |
| Yes | 33 (40.7%) | 44 (54.3%) | | 0.084 |
| No | 48 (59.3%) | 37 (45.7%) | | |

| | Age Group | | | | *p*-Value |
| --- | --- | --- | --- | --- | --- |
| | 18 to 20 | 21 to 25 | 26 to 30 | Above 31 | |
| Yes | 37 (55.2%) | 33 (44.6%) | 4 (36.4%) | 3 (30%) | 0.305 |
| No | 30 (44.8%) | 41 (55.4%) | 7 (63.6%) | 7 (70%) | |

| | Monthly Income | | | | | *p*-Value |
| --- | --- | --- | --- | --- | --- | --- |
| | No monthly income | <1 minimum wage | 1 to 3 minimum wages | >3 to 5 minimum wages | >5 minimum wages | |
| Yes | 5 (45.5%) | 46 (47.4%) | 21 (47.7%) | 3 (42.9%) | 1 (33.3%) | 0.971 |
| No | 6 (54.5%) | 51 (52.6%) | 23 (52.3%) | 4 (57.1%) | 2 (66.7%) | |

The frequency of adoption of health and well-being behaviors promoted by DIs is also higher for younger students (between 18 and 20 years old). In older age groups, there is a tendency for the adoption of these behaviors to decrease. However, there was no significant association. Monthly income did not appear to be a decisive factor for this aspect of behavior adoption. Very similar percentages are observed, except for college students who have an income above five minimum wages.

Table 2 presents the absolute and relative frequency values of responses related to the use of Instagram according to gender. It is possible to observe, initially, that the profile of the public investigated is composed, in its majority, of young people aged 18 to 20 and 21

to 25, with low monthly income. However, the male public has a higher income (one to three minimum wages) than the women (one minimum wage).

**Table 2.** Simple frequency and percentage values of responses related to the use of Instagram according to gender.

| Variables | Female | | Male | | *p*-Value |
|---|---|---|---|---|---|
| | Freq. | % | Freq. | % | |
| Age group | | | | | |
| 18 to 20 | 32 | 39.5 | 35 | 43.2 | |
| 21 to 25 | 37 | 45.7 | 37 | 45.7 | |
| 26 to 30 | 8 | 9.9 | 3 | 3.7 | 0.422 |
| >30 | 4 | 4.9 | 6 | 7.4 | |
| Monthly income | | | | | |
| No monthly income | 4 | 4.9 | 7 | 8.6 | |
| <1 minimum wage | 57 | 70.4 | 40 | 49.4 | |
| 1 to 3 minimum wages | 14 | 17.3 | 30 | 37 | 0.039 * |
| >3 to 5 minimum wages | 4 | 4.9 | 3 | 3.7 | |
| >5 minimum wages | 2 | 2.5 | 1 | 1.2 | |
| How often do you use Instagram and the content posted by Digital Influencers to learn about health and well-being? (Frequency) | | | | | |
| Always | 04 | 5.0 | 03 | 3.7 | |
| Frequently | 29 | 35.8 | 25 | 30.9 | |
| Rarely | 39 | 48.1 | 42 | 51.9 | 0.860 |
| Never | 09 | 11.1 | 11 | 13.5 | |
| Do you adopt the health and wellness behavior posted on Digital Influencers' Instagram into your life? (Adoption of Behaviors) | | | | | |
| Yes | 33 | 40.7 | 44 | 54.3 | |
| No | 48 | 59.3 | 37 | 45.7 | 0.084 |
| Do you consider that the Digital Influencers you follow on Instagram present a body standard that is attainable by everyone who follows their suggestions and/or adopts their behaviors? (Body Standard) | | | | | |
| Yes | 17 | 21.0 | 29 | 35.8 | |
| No | 64 | 79.0 | 52 | 64.2 | 0.037 * |
| Do you consider that the Digital Influencers you follow on Instagram present a standard of health attainable by all who follow their suggestions and/or adopt their behaviors? (Health Standard) | | | | | |
| Yes | 25 | 30.9 | 21 | 25.9 | |
| No | 56 | 69.1 | 60 | 74.1 | 0.486 |
| Do you consider that the contents disclosed by Digital Influencers that deals with health and well-being contribute to your self-care? (Self-care) | | | | | |
| Always | 07 | 8.6 | 12 | 14.9 | |
| Frequently | 34 | 42.0 | 39 | 48.1 | |
| Rarely | 37 | 45.7 | 27 | 33.3 | 0.363 |
| Never | 03 | 3.7 | 03 | 3.7 | |

* Chi-square test—*p* < 0.05.

Regarding the digital influencers' body standards (related to the question "Do you consider that the Digital Influencers you follow on Instagram present a body standard that is attainable by everyone who follows their suggestions and/or adopts their behaviors?"), there were significant values when associated with the gender variable: 79% of women and 64.2% of men do not consider it possible to reach the DI body standards by following the suggestions and behaviors they divulge (*p* = 0.037).

*3.2. Network Analysis: Association between Instagram Use Profile and the Adoption of Behaviors Related to Health and Well-Being*

An analysis of artificial networks was applied to verify the associations between the variables that characterize the use of Instagram for the adoption of health and well-being behaviors and income, age group, and the semester the students are attending. Figure 1 shows that, for both men and women, the adoption of health and well-being behaviors promoted by DIs (node 6) presents positive and strong associations with the notion that health and well-being content promoted by DIs contributes to the self-care of the individuals in this study (node 9) (0.824 for men and 0.815 for woman).

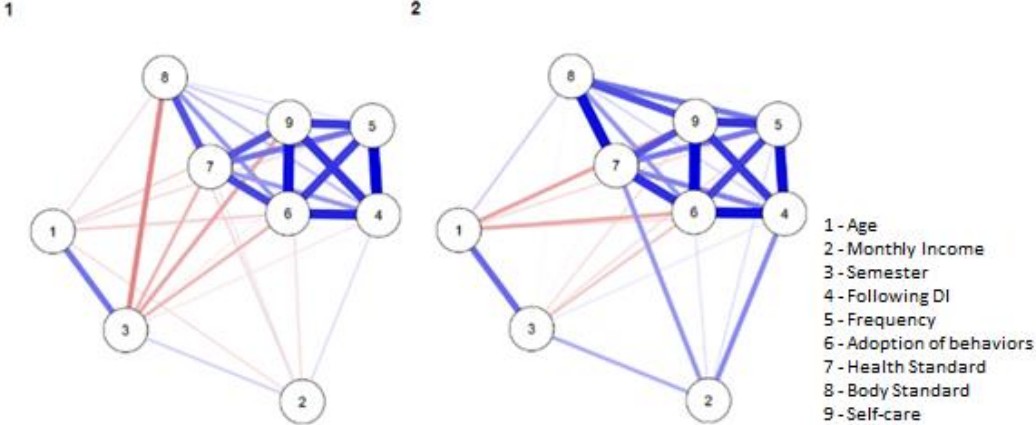

**Figure 1.** Network analysis of the associations between Instagram use profile as a means for the adoption of behaviors related to health, age group, monthly income, and semester for men (1) and women (2). Legend: Following DI—following digital influencer; Frequency—frequency of access to Instagram; Adoption of behaviors—whether the individual adheres to the behaviors suggested by the digital influencer; Health Standard.

Positive associations can also be observed among nodes 4 (following DI), 5 (frequency of accessing Instagram to search for health and well-being content), and 6 (adoption of these behaviors) for both genders. Moreover, for women, the network analysis graphs (Figure 1) show that the relationships between the variables that characterize the use profile and the sociodemographic variables, such as age (0.162 for men and 0.269 for woman) and monthly income (0.031 for men and 0.091), are stronger in comparison to men, but still weak. Regarding age, the association is negative, that is, the greater the women's age, the lower the adoption of healthy habits suggested by DIs (node 6).

For women, nodes 8 (believes the DIs they follow on Instagram present a body standard that can be achieved by everyone who follows their suggestions and adopts their behaviors) and 5 (frequency of use of Instagram and DIs to obtain information about health and well-being) are more strongly related (0.429) in comparison to men (0.098), which may indicate that the greater the consumption of this content, the greater the belief that this body standard can be achieved by them.

For men, the semester has negative associations with belief in achieving the body standard of DIs (node 8) (−0.408). This node also has a stronger negative relationship with nodes 6 (adoption of health and well-being behaviors promoted by DIs) (−0.240), 7 (belief that the health standards of the DIs can be achieved) (−0.248), 8 (belief that the body standards of the DIs can be achieved) (0.408), and 9 (the health and well-being content produced by DIs contributes to the self-care of the individuals under study) (0.298). Node 8 is the one with the strongest connection relationship. The relationships between the semester and the profile of using Instagram to obtain information about health and well-being are much lower for women (see Table 3).

**Table 3.** Centrality measures of the networks.

| Variables | Betweenness | | Closeness | | Strength | |
|---|---|---|---|---|---|---|
| | **Male** | **Female** | **Male** | **Female** | **Male** | **Female** |
| Age | −0.868 | 0.632 | −1.157 | −1.039 | −1.241 | −1.097 |
| Monthly income | −0.868 | 0.000 | −2.135 | −0.648 | −1.674 | −1.341 |
| Semester | 2.062 | −0.632 | 0.253 | −1.736 | −0.493 | −1.298 |
| Following DI | 0.109 | 0.632 | 0.117 | 0.468 | 0.654 | 0.649 |
| Frequency | −0.868 | −1.265 | 0.259 | 0.352 | 0.497 | 0.596 |
| Adoption of behavior | −0.380 | 0.000 | 0.724 | 1.092 | 1.154 | 1.089 |
| Health standard | 0.109 | 0.000 | 0.624 | 0.465 | 0.500 | 0.568 |
| Body standard | −0.380 | −1.265 | 0.361 | −0.249 | −0.396 | −0.166 |
| Self-care | 1.085 | 1.897 | 0.955 | 1.294 | 0.999 | 1.000 |

Table 3 demonstrates that measures of centrality show similar results for men and women regarding closeness and strength. Node 9 is the variable with greater closeness among the other nodes. Regarding the strength of the correlations between the nodes of the network, node 6 reveals the highest indices.

On the other hand, in the measure of betweenness, which represents the intermediation ability of a variable with different pairs of nodes in the network, there was a difference between the genders. In women, node 9 (the health and well-being content produced by DIs contributes to the self-care of the individuals under study) represented the greatest influence, suggesting that the belief that such content contributes to the self-care of the participants establishes relationships with other pairs of variables, such as frequency of use and the adoption of behaviors promoted by DIs.

For men, the semester (2.062) they were attending (node 3) was the node with greater betweenness, indicating that the association between the different pairs of nodes is mediated by the semester these physical education students were attending.

Monthly income demonstrated greater association with following DIs, among women (0.365), compared to men (0.117) (Tables 4 and 5). For women, node 2 (monthly income) had a stronger relationship with nodes 4 (following DIs who produce health and well-being content on Instagram) and 7 (belief that the health standards of the DIs can be achieved). Thus, the higher the monthly income, the more frequent the consumption of this content.

**Table 4.** Weight matrix for the associations between the network variables for men.

| Variables | 1 | 2 | 3 | 4 | 5 | 6 | 7 | 8 | 9 |
|---|---|---|---|---|---|---|---|---|---|
| 1. Age | - | | | | | | | | |
| 2. Monthly income | −0.109 | - | | | | | | | |
| 3. Semester | 0.472 | 0.158 | - | | | | | | |
| 4. Following DI | −0.055 | 0.117 | −0.081 | - | | | | | |
| 5. Frequency | −0.124 | 0.005 | −0.062 | 0.880 | - | | | | |
| 6. Adoption of behavior | −0.162 | −0.031 | −0.240 | 0.759 | 0.665 | - | | | |
| 7. Health standard | −0.022 | 0.087 | −0.248 | 0.349 | 0.474 | 0.632 | - | | |
| 8. Body standard | −0.114 | −0.129 | −0.408 | 0.207 | 0.098 | 0.352 | 0.591 | - | |
| 9. Self-care | −0.137 | −0.112 | −0.298 | 0.702 | 0.678 | 0.824 | 0.588 | 0.167 | - |

**Table 5.** Weight matrix for the associations between the network variables for women.

| Variables | 1 | 2 | 3 | 4 | 5 | 6 | 7 | 8 | 9 |
|---|---|---|---|---|---|---|---|---|---|
| 1. Age | - | | | | | | | | |
| 2. Monthly income | −0.015 | - | | | | | | | |
| 3. Semester | 0.471 | 0.251 | - | | | | | | |
| 4. Following DI | −0.110 | 0.365 | 0.074 | - | | | | | |
| 5. Frequency | −0.113 | 0.100 | −0.105 | 0.775 | - | | | | |
| 6. Adoption of behavior | −0.269 | 0.091 | −0.173 | 0.774 | 0.598 | - | | | |
| 7. Health standard | −0.024 | 0.325 | −0.026 | 0.390 | 0.422 | 0.658 | - | | |
| 8. Body standard | 0.150 | −0.018 | −0.021 | 0.128 | 0.429 | 0.327 | 0.767 | - | |
| 9. Self-care | −0.280 | −0.016 | −0.105 | 0.632 | 0.650 | 0.815 | 0.550 | 0.562 | - |

## 4. Discussion

The results presented findings in line with the literature regarding the differences [40,42,69] and similarities [70] in the use that female and male audiences make of Instagram and the health and well-being content produced by digital influencers.

The analysis of descriptive statistics allowed us to better understand the frequency of certain behaviors and its relation to gender and other variables. In the network analysis, the strength of the relationships between the variables allows for a better differentiation between the genders.

In general, Internet use is associated with demographic aspects, mainly age, education, and gender [71]. Regarding Instagram, research [42] has shown that the relationship regarding the use of this SN is basically the same between the genders, including when comparing education and monthly income, although use by women is slightly higher than that of men.

Other studies argue that gender, in general, presents greater differences when associated with motivations and modes of use for each type of population, which is supported by our findings [69]. For example, for women, content that contributes to self-care has greater influence, suggesting greater importance.

In the statistical analysis, we found that the frequency of adoption of behaviors (node 3) is higher for men than for women. However, research indicates that use by men is more related to entertainment, games, and technology [40]. This new factor may add to the literature on yet another form of use by male Instagram users.

Regarding the frequency of consumption of health and well-being content by men, few studies prove cause-and-effect relationships for this segment of the population, focusing more on the female body, with the frequency of Instagram access being led by women [40].

Regarding the consumption of such content, some studies argue that it can cause both body dissatisfaction through a process of self-comparison and the internalization of an ideal of a muscular body (in the case of men), as well as contribute to self-care and positive changes in behavior [11,71].

However, as it was possible to observe, our study demonstrates that the issue of body standards needs further investigation, since both men and women understand that it is not possible to achieve the body standard of digital influencers from the suggestions disseminated by them, providing the understanding that the students investigated understand the limits of interaction with body-related content that can cause negative effects.

In other studies, gender revealed no differences in terms of Instagram use and body concerns. Both men and women were equally vulnerable to body image problems related to the use of the platform and self-comparison [70].

Another factor in the literature was the greater presence of young people. Studies [42] have shown that age is related to the use of the platform, that is, the older the individual, the lower their use of SNs. Regarding Instagram, the presence of younger groups is explained by the fact that it is a photo posting tool, which proves to be more attractive to this segment of the population.

Another explanation for this fact is that the young public has more experience and interaction with SNs due to the technological environment in which they grew up [2,72]. Thus, age is an important factor in the mediation of Internet activities, since young people use services for the purpose of seeking information, entertainment, and social interactions [73]. This trend also emerges with the intense use of Instagram by young students of both genders [74].

Regarding monthly income, our study shows that men have a higher income than women, however, this variable only showed significant differences for women. Data demonstrate monthly income as an impact factor for the use of SNs such as Instagram, to the extent that the higher the income, the greater the use, due to time availability or for work-oriented uses [42].

As indicated in the results, the graphs of the network analysis show that, for women, the relationships are stronger between the variables that characterize the use profile and the sociodemographic variables, such as age and monthly income.

This aspect is supported by the associations between nodes 6 (adoption of health and well-being behaviors) and 9 (the health and well-being content produced by the DIs contributes to the self-care of the individuals under study), which exhibit a slightly stronger relationship for the women. In this case, the understanding that content related to health and well-being contributes to their self-care may be more strongly associated with the adoption of these health behaviors among the women in this study. Based on the results, age is an influential factor for understanding what would be useful for their self-care. In this case, adopting behaviors focused on self-care and not on the pursuit of a certain body type can represent a positive factor and one of the characteristics of using Instagram to inform oneself about health and well-being.

Body type is a much-discussed topic and, in this case, it is related to posts of images of thin bodies in situations of exercise, leisure, food, and other aspects that are associated with healthy behaviors, leading many users to believe that a specific body type is associated with health [1,12,75,76].

This issue is essential for women, as it is widely discussed that this audience tends to suffer greater negative consequences related to body image, which may cause anxiety, depression, and body surveillance, among others [32,35,75,77], due to the more passive use of SNs [31] and the consumption of content aimed at the fitness lifestyle.

Passive use, high investment (time, frequency of use), and ostracism on SNs were identified as factors negatively related to subjective well-being, causing depression, bad mood, body and life dissatisfaction, as well as low self-esteem [29].

Research indicates that a lot of content contains information with a strong capacity to contribute positively to changes in health behaviors. On the other hand, the information and the way it is conveyed on Instagram (followed by photos associating health and well-being with a young and slender body, certain types of food and routines) contribute to the dissemination of harmful health behaviors, especially if accompanied by uncritical (or passive) consumption of this content [1,29,30,33,75,77]. Conversely, active use is more related to the search for information, and it is influenced by age and social status, that is, the older the individual and the higher their socioeconomic status, the more participatory the use of the social network becomes, among women [40].

Overall, socioeconomic status, gender, and age are variables that influence SN behavior in young people, including the intense search for information about diet, exercise, and well-being [39,40]. Thus, research with sociodemographic variables in different contexts is necessary to understand such relationships with the use of the Internet and SNs.

Even though they exhibit a slightly stronger relationship among women, nodes 9 (closeness) and 6 (strength) were represented among both genders, such that content that contributes to self-care and behavior adoption is the most central issue for men and women when it comes to using Instagram to inform themselves about health and well-being.

This result may suggest that trust in the content produced by DIs on Instagram about health and well-being is representative. Node 6, about the adoption of behaviors, appears

as the most powerful variable for both genders, corroborating the idea that trust in DIs is fundamental to adopting such behaviors.

Digital influencers can be regarded as individuals who use their social power with an objective, generally related to consumption, and this social power, together with this commercial objective, is highly significant, especially in engaging others to consume, providing motivation for certain attitudes and modulating public opinion. In this case, the DIs and micro-celebrities (who have influence) become commercial tools, who extend beyond the limits of private life and use their forms of communication as a means of engaging others to consume [78].

In this regard, it is important to say that a DI can be a reference for men and women with regard to health information when it is searched for in an SN. Many types of content focus on the promotion of a healthy lifestyle. Fitspiration, for example, is an influential movement widely discussed in the scientific literature as a phenomenon, found on social media, which disseminates images with the goal of inspiring people to adhere to a diet and exercise for a healthy life [75].

By becoming a reference for information on health and well-being, DIs will reveal their points of view based on their routines, which may cause users to fixate on a certain lifestyle, leading to compulsive and obsessive processes around reductive patterns and conceptions of health [9,12,26,40].

In the meantime, it is important to emphasize that body dissatisfaction can result from this dialogue between a body ideal and the intangibility of its achievement by everyone. It is an aspect quite present during adolescence and youth, especially in women, due to strong Western pressure on the female body. With the dissemination of fitness profiles and the pressure of achieving certain body standards, and with a focus on image and a healthy lifestyle, this dissatisfaction tends to increase [21].

Finally, betweenness exhibits differences between the genders. This centrality measure represents the strength of influence between the variables. For men, it was the semester (node 3), whereas, for women, it was that content published by DIs on health and well-being contributing to self-care (node 9). This indicates that, for men, education is a means of influencing the other variables in this study and, for women, it is the content itself and the possibility it may contribute to their self-care.

Research indicates that, overall, younger men with a higher level of education make more frequent and varied uses of the Internet [71]. Education, especially higher education, is an impact factor: the higher the educational level, the greater the presence on SNs, which may be related to a search for information on SNs [42].

Based on these results, it is suggested that men believe less in standards presented by DIs than women, when taking into consideration their education (semester of the course they are attending). Thus, the more advanced their education, the lower their belief that DIs disseminate body and health standards that are possible to achieve. This is also related to the adoption of health behaviors and the understanding that this type of content contributes to self-care.

Furthermore, education emerges as an expressive variable to mediate the search for information on the Internet, mainly related to health and well-being issues and the possible consequences related to the types of information available. College students lead this field in using the Internet and SNs to search for information. These differences are also linked to the context of the country in which they are investigated [73].

Therefore, initiatives on education should aim at a more active approach to the use of SNs and educational interventions for more passive users [39]. Moreover, the tools offered by the different SN platforms must be examined in order to be used towards a positive change in health behaviors, guided by responsible professionals [14].

## 5. Conclusions

Overall, the findings indicate that the profile of using Instagram to obtain information about health and well-being is influenced by sociodemographic variables in general, such

as gender, age, income, and semester, among college students. Gender presented some differences in the use, but, in general, both men and women adopt the suggestions present in the health and well-being content produced by digital influencers. We found that, although the frequency of adoption of behaviors is higher for men, women have stronger relationships with the variables of the profile of using Instagram to obtain information about health.

In this case, it is not the frequency of adoption of these behaviors that affects women, but the stronger relationship with the other variables. This fact is interesting and exemplifies the importance of using the technique of artificial network analysis to examine the relationship between sociodemographic variables and use profile on Instagram, a result that was not found by descriptive statistics only.

Therefore, further actions are necessary to better examine the impact, for both genders, of the adoption of this content. Although the literature already provides some information, it is still necessary to explore the diversity of existing contexts. With this in mind, we suggest the development of educational actions that allow a more critical exploration of college students from this context and individuals from other contexts, as well as a critical use of the findings in order to minimize any negative effects resulting from the passive use of this content.

## 6. Study Limitations

This work has some limitations, and the results should be interpreted taking those limitations into account. The first limitation is related to the convenience sample. The distribution by semester and age group was not the same, since participation in the study was voluntary and the pandemic also influenced the number of students attending the classes at the time of data collection. Furthermore, the analysis is based on a set of quantitative data, which does not allow for a deeper understanding of the relationships between the variables, and the individuals under study are students of a course in the health field, which may have influenced the choices of content that they view on their social media. Further research should include a more homogeneous sample distribution and mixed methods that seek complementarity, as well as examine different population contexts.

**Supplementary Materials:** The following supporting information can be downloaded at: https://www.mdpi.com/article/10.3390/soc13020045/s1, Supplementary Materials: Questionnaire.

**Author Contributions:** K.P.: Conceptualization, Methodology, Data curation, Writing—original draft, Investigation, Writing—review & editing; C.L.A.C.: Methodology, Writing—original draft, review & editing; A.C.C.: Writing—review & editing; I.R.: Supervision, Validation, Visualization, Writing—review & editing; A.B.: Supervision, Validation, Visualization, Writing—review & editing. All authors have read and agreed to the published version of the manuscript.

**Funding:** Partial support for this work was provided by the Federal Institute of Education Science and Technology of Ceará, Limoeiro do Norte campus. This work was partially supported by the Portuguese Government, through the Foundation for Science and Technology, IP (FCT), under the multi-year funding awarded to CIIE (grants no. UIDB/00167/2020 and UIDP/00167/2020).

**Institutional Review Board Statement:** The study received ethical approval from the Ethics Commission of the University of Trás-os-Montes and Alto Douro (Doc89-CE-UTAD-2020) and Consubstantiated Opinion from State University of Rio Grande do Norte, Brazil (42856720.0.0000.5294).

**Informed Consent Statement:** Informed consent was obtained from all subjects involved in the study.

**Data Availability Statement:** The complete dataset is still under analysis and will be released in future.

**Conflicts of Interest:** The authors declare no conflict of interest.

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
