# Peer review of "Use of Instagram as a Resource for the Adoption of Behaviors Related to Health and Well-Being of Young College Students: Associations between Use Profile and Sociodemographic Variables—A Cross-Sectional Study"

_societies, doi:10.3390/soc13020045_

Round 1

Reviewer 1 Report (Previous Reviewer 1)

The authors revealed the ability to give a positive response to scientific arbitration, which is markedly positive. The changes introduced substantially enhance the article and give rise to possible publication of the work.

Author Response

Thank you for your constructive comments. 

Reviewer 2 Report (Previous Reviewer 2)

The Authors have improved the manuscript considerably. However, there are still issues limiting the value of their input.

The description of the study aims is rather convoluted.

The process of the validation of the questionnaire is unclear. Strong emphasis on the role of academic staff in the process is not fully explained in relation to peer validation and pre-testing. It is hardly understandable why ‘professors’ are supposed to participate in this part (‘peer validation” or pre-testing) of the instrument development.

The questionnaire used in the study should be provided as supplementary material.

What is the significance of relationships between variables included in network analysis?

Significant parts of the Conclusions section sound more like the Discussion, e.g., 666-677.

The manuscript should be once again checked by a native speaker. It seems the Authors overuse capital letters for some terms, e.g., when addressing the names of subjects or describing the options of the variables (lines 296-303).

It is not clear why the title of the paper is capitalized.

Some sentences are awkward.

There are some problems with table 3; overlapping rows, headings of the columns not aligned with values in columns.

Why are 0s needed in tables 3 and 4?

Tables 3 and 4 – it is not clear at once what numerical values 1-9 stand for (maybe the names of the variables in the first columns should be preceded with the appropriate numbers, e.g., 9. Self-Care or 8. Body standard.

Author Response

  1. The Authors have improved the manuscript considerably. However, there are still issues limiting the value of their input.

A Thank you for your constructive comments.

  1. The description of the study aims is rather convoluted.

A Thank you for your constructive comments. We do not understand this comment. Could you be more specific? We tried to describe the study with 3 objectives: The first one is general. The other two aimed to split the study into two stages in which it was analyzed: sociodemographic profile and usage behaviors. We hope that this explanation helps to clarify. But we can delete the last two objectives.

  1. The process of the validation of the questionnaire is unclear. Strong emphasis on the role of academic staff in the process is not fully explained in relation to peer validation and pre-testing. It is hardly understandable why ‘professors’ are supposed to participate in this part (‘peer validation” or pre-testing) of the instrument development.

A Thank you for your constructive comments. Concerning the validation process, we rely on a framework that instructs us to submit the questionnaire to 3 groups: peer colleagues, to find out if the questions answer the purpose of the study; subject matter experts, to prevent the inclusion of obvious questions that may occur during the design process; and target respondents, to understand, from the point of view of the public with characteristics close to the target group, if the study is understandable. These steps help verify the clarity of the instructions, questions, and answers, and whether the scales and the contents of the answers are as expected.

  1. The questionnaire used in the study should be provided as supplementary material.

A Thank you for your constructive comments. We included the questionnaire as supplementary material.

  1. What is the significance of relationships between variables included in network analysis?

A Thank you for your constructive comments. The purpose of using neural networks in the study was to use topological analysis to understand the relationships and dynamics between variables. Not to generate a predictive model. Furthermore, the weight matrix provided by the software (JASP) used does not provide p-value.

  1. Significant parts of the Conclusions section sound more like the Discussion, e.g., 666-677.

A Thank you for your constructive comments. This information was deleted.

  1. The manuscript should be once again checked by a native speaker. It seems the Authors overuse capital letters for some terms, e.g., when addressing the names of subjects or describing the options of the variables (lines 296-303).

A Thank you for your constructive comments. The manuscript was revised by a native speaker.

  1. It is not clear why the title of the paper is capitalized.

A Thank you for your constructive comments. The title was corrected.

  1. Some sentences are awkward.

A Thank you for your constructive comments. The sentences were revised by a native speaker.

  1. There are some problems with table 3; overlapping rows, headings of the columns not aligned with values in columns.

A Thank you for your constructive comments. We correct the tables in the manuscript.

  1. Why are 0s needed in tables 3 and 4?

A Thank you for your constructive comments. We already changed.

  1. Tables 3 and 4 – it is not clear at once what numerical values 1-9 stand for (maybe the names of the variables in the first columns should be preceded with the appropriate numbers, e.g., 9. Self-Care or 8. Body standard.

A Thank you for your constructive comments. We already changed.

Round 2

Reviewer 2 Report (Previous Reviewer 2)

The Authors provided further explanations and have amended some elements indicated in the review. The paper has still some weaknesses, but I believe it can be published in current version.

This manuscript is a resubmission of an earlier submission. The following is a list of the peer review reports and author responses from that submission.

Round 1

Reviewer 1 Report

The design of this study is not a “case study”. It is a descriptive, servey design. In fact, the definition of the objectives and the starting question are evidence that a case was not studied: “…prevalence of Instagram use as a resource for the adoption of behaviors related to health and well-being, among college students”.

How the sample of 162 students was selected must be indicated. Instrument: in this section you must report the dimensions / sub-dimensions of the instrument (which are mentioned), the objectives of each section / subsection, the questions, indicators, regulations / fundamental authors. In addition to the typology of questions, the number of questions for each section and how (types and by whom) the instrument was validated should be reported, as well as the changes that occurred as a result of this validation.

If a pre-test was carried out, it is important to give an account of the reliability index and how to act accordingly. What is written in this section is incipient. Was the questionnaire applied in class or on social media?

Salary: who receives 3 salaries, chooses 1 to 3 salaries or 3 to 5 salaries? It is assumed that the question asked is not unique. The same applies with class 3 to 5 salaries and 5 salaries. The network analysis indicates some questions: that is why it would be important in the instrument to report the questions.

We do not need to know more than once that this study is part of a PhD project.

It is not relevant to the article or the reading public.

Conclusions: appear as generalizable, but cannot be. The study design is not a case study. It is a survey type study. However, the sample is small (N = 162) and does not allow generalizations. The reliability of the information collection instrument is unknown, which is crucial. Participation in a study is always voluntary!

“Furthermore, the analysis is based on a set of quantitative data, which does not allow for a deeper understanding of the relationships between the variables, and the individuals under study are students of a course in the health field, which can influence the choices of content that they will view on their social media”. Now, precisely because of these factors, it is not a case study and the results cannot be generalized to “university students” when they apply only to the sample (which is not representative of the university population in Brazil).

It is not understood how “individuals under study are students of a course in the health field” are assigned to the higher education course Physical Education.

It is considered that the work is not in a position to make a relevant contribution to the scientific community, even though the topic is very relevant. A general review of the methods and the way in which conclusions are drawn is suggested.

References must strictly follow the rules of the Societies journal (e.g., titles of books and journals in italics, use of p., etc.).

Reviewer 2 Report

I regret that this manuscript suffers from so many deficiencies and serious failures that I cannot even recommend revisions. It is not clear in linguistic aspect, it does not conform to scientific language, and basically, it looks like the Authors should read more papers reporting the results of cross-sectional studies and appropriate analytical methods. It is prepared with high carelessness, without respect for formatting rules required by the journal. The list of references is cut at some moment, and it is unclear what sources the Authors refer to in the Discussion. The Methods are not clearly and transparently reported; there are deficiencies in the description of methods and a lack of appropriate vocabulary. The Results, as most of the manuscript, are presented in a careless and not transparent way.

The Introduction should be considerably improved and extended. It is rather shallow, and key aspects like health and wellbeing, the role of social media influencers, the use of social media in health-related social campaigns, and many others are very weakly presented. Overall, this paper's conceptual and taxonomic framework is hardly visible. It is difficult to see what the Authors want to do as they use key terms in rather popular meanings without striving for a well-defined analytic space.

As Author focused on the concepts of health and wellbeing, I would suggest wider development on understanding health. Contradicting health as a lack of diseases to the traditional 70 years old definition from WHO sounds like a rather obsolete practice. I would recommend, especially in the context of new media and the overwhelming model of multidirectional communication, to discuss more health as the strategy of adaptation to challenges and dynamic conditions. The requirement to follow health as a state of complete multilevel wellbeing results in the situation that hardly anybody is really healthy.

Key concepts and definitions about health behaviors should be provided to understand what the Authors understand by this term.

The motive of the role of lifestyle influences and Internet/social media influencers, in general, is practically not covered. This topic requires a more significant description and systematic presentation, considering the scope of the analysis.

The importance of social media in health-related campaigns is another aspect covered very poorly.

In the preceding paragraph of the Introduction (lines 101-102), the Authors write about a research group on cultural differences in using social media, but actually, their study is focused only on sociodemographic variables. Overall, the objective is not clearly presented and justified. It is unclear why simple sociodemographic variables are so important when assessing Instagram's role in shaping young people's health behaviors.

Furthermore, it is not clear why college students and not other populations were included in the study. It is also unclear what type of content on Instagram is addressed in the study. This should be somehow specific as the content of social media is decidedly of various character, and we do not know if the Authors assess particular types of influencers/content/campaigns. It is a highly generic and unsatisfactory approach.

Why should the role of Instagram be analyzed in a very specific group of college students, especially since they study physical education? This group does not seem to be the best for understanding the role of the health-related content on Instagram and anticipate the finding to the whole population of youth or young adults. The study is biased as the education process specifically directs the interest of the study group and their perception of the content from the Internet and other sources.

Finally, no hypotheses are formulated.

Methods

Although the Authors do not present key concepts and definitions vital for the analytical framework, they tend to explain obvious terms like ‘probabilistic sample.’ It seems that such definitions are not necessary for the peer-reviewed journal as such.

Was the ideal gender balance in the study group a result of the Authors’ approach?

In the subsection  ‘Instrument,’ the text mentions “the researcher” in one place, but then it writes “we considered.” So is this the research conducted by one or many researchers? The text should be kept consistent.

The text about ‘instrument’ contains many general statements, but we do not know much about the questionnaire. For example, no mention of the number of items, items in sections, scale (e.g., Likert) to respond to items, or the use of a standardized tool (or not).

Overall, the text of this subsection is not fully understood, and it seems that some words are not used properly, e.g., ‘sessions,’ ‘contemplate.’

It is unclear what the Authors mean by being “validated by a panel of experts’. The procedure is described in a very vague way. What are the people involved in so-called ‘validation’? Why were they chosen? What is their experience in such work?

To what extent ‘Ethics Committees’ may be responsible for validation? Why ‘Committees’ and not ‘Committee’? Why was ‘pre-testing’ conducted?

There is a huge number of doubts and questions here, and they are not answered clearly. Considering all these problems, one cannot be certain about the quality of the instrument, the study, and the resulting analysis.

The fact that the study sample is practically identical to the participants of the same courses is a substantial weakness of the stud and a potential source of significant bias.

The description of what the Authors call ‘analysis procedures’ (I strongly recommend reaching for professional linguistic services and other papers of this type) is rather confusing and convoluted, starting from the first paragraph about descriptive statistics.

The use of neural networks requires clear presentation; now, it is unclear how this specific approach was applied. Why was NN analysis applied and not regression modeling? What is the advantage of NN over other techniques, considering a relatively limited number of variables?

The variables used in the analysis are not explained. One cannot understand what ‘1 salary’, ‘2 salaries’ etc. mean. It is not even clear what health behavior was analyzed in the study.

Results

Overall, the presentation of results is strikingly poor and non-transparent. It isn't very careful. The headings of the table are displaced far from the tables; the headings do not provide sufficient information about the content of the tables. When including the variables in the tables, the normal names of variables should be provided, not some strange structures like Health-Standard-DI. The Authors should also take care to distinguish subsections and logical parts of the analysis.

Table 1 is overwhelming straightforward, and awkward. Why were only sex, age, and ‘salary’ (whatever it means) included? The characteristics of the study sample should be presented here with continuous variables described with appropriate descriptive statistics and categorical variables in the table showing the categories for each variable with absolute and relative frequencies, including health behaviors and variables characterizing the use of Instagram and potentially other variables used in the study.

The results of network analysis should be described in terms of the parameters of the models and not only in a narrative way (referring to figure 1)

Table 2 – the heading of the table should be more informative; there is no information why some values were bolded. There is no need to provide the source in case of the authors of the paper

Discussion

It is difficult to understand this paper's justification and scope of analysis; this leads to the situation that a reader hardly knows what the Authors discuss. Apart from this, once again, a symptom of careless and neglected manuscript preparation, a significant number of references is not provided in the reference list.

As the Authors do not use fully understandable language to describe statistical interrelations, it is rather difficult to grasp the strategy behind the presentation of the statements in the Discussion.

The paper suffers from many limitations, but even they should be described in an understandable, explicit way. Furthermore, they should be included at the end of the Discussion and not after the Conclusions.

Conclusions

The Conclusions are not fully clear; the first paragraph is internally contradictory, is or is not the use of Instagram influenced by sex?

References

It looks like the numbering of the references is repeated; why? The formatting of the references does not adhere to the style required by the MDPI. The list of references is inconsistent with the references in the text. Apparently, some references are missing. It is not clear if the references in the text are appropriate.

I would recommend that the Authors read a bunch of papers reporting the results of cross-sectional studies employing the survey technique, as now the terminology and description of the subsection required in the original research are highly unsatisfactory and inadequate. Reaching for a relevant STROBE checklist would also be needed.

It is unclear why the Authors use capital letters in such a specific way, e.g. ‘between Education, use Instagram, and the search for Digital Influencers.’

The help of professional linguistic services is necessary for this paper.